# Assumption Questioning: Latent Copying and Reward Exploitation in Question Generation

## Abstract

Question generation is an important task for improving our ability to process natural language data, with additional challenges over other sequence transformation tasks. Recent approaches use modifications to a Seq2Seq architecture inspired by advances in machine translation, but unlike translation the input and output vocabularies overlap significantly, and there are many different valid questions for each input. Approaches using copy mechanisms and reinforcement learning have shown promising results, but there are ambiguities in the exact implementation that have not yet been investigated. We show that by removing inductive bias from the model and allowing the choice of generation path to become latent, we achieve substantial improvements over implementations biased with both naive and smart heuristics. We perform a human evaluation to confirm these findings. We show that although policy gradient methods may be used to decouple training from the ground truth and optimise directly for quality metrics that have previously been assumed to be good choices, these objectives are poorly aligned with human judgement and the model simply learns to exploit the weaknesses of the reward source. Finally, we show that an adversarial objective learned directly from the ground truth data is not able to generate a useful training signal.

## 1 Introduction

Posing questions about a document in natural language is a crucial aspect of the effort to automatically process natural language data, enabling machines to ask clarification questions (Saeidi et al., 2018), become more robust to queries (Yu et al., 2018), and to act as automatic tutors (Heilman & Smith, 2010).

While questions often have a unique answer, the inverse is rarely true; there are multiple ways of phrasing a question, and there can be semantically different questions with the same answer. The ability to generate questions therefore provides a mechanism for augmenting existing question answering datasets or automatically annotating new ones, improving the resilience of question answering models.

Recent approaches to question generation or *answer questioning* (AQ) have used Seq2Seq (Sutskever et al., 2014) models with attention (Bahdanau et al., 2014) and a form of copy mechanism (Vinyals et al., 2015; Gulcehre et al., 2016). Such models are trained to generate a plausible question, conditioned on an input document and answer span within that document (Zhou et al., 2018; Du et al., 2017; Du & Cardie, 2018; Yuan et al., 2017).

Many innovations in sequence generation have been led by a motivation to improve neural machine translation (NMT) models, where the input and output vocabularies are largely orthogonal and the correct lengths of the input and output sequences are often similar. Shi et al. (2016) showed that specific units within NMT models learn to count the sequence length during decoding, to help achieve the correct output length. In this way, NMT can be viewed as separate alignment and per-token transformation tasks.

There is no particular intuition that the lengths of a context document and a question about that document should be related; there is a correlation of only 0.009 between the lengths of contexts and related questions in the popular SQuAD question answering dataset (Rajpurkar et al., 2016). The document and the question are also written in the same language, and so use overlapping vocabu-

laries; 88% of the words from all questions in the training set also appear in at least one context document. AQ therefore presents significantly different challenges to NMT.

Recent works on AQ have used various formulations of copy mechanisms, but have not investigated which approach should be preferred. We show that by removing inductive bias from the model and allowing the choice of generation path to become latent, we achieve substantial improvements over both a naive and more principled biased implementation.

There are currently no dedicated question generation datasets, and recent work has used the context-question-answer triples available in SQuAD. Only a single question is available for each context-answer pair, and models are trained using teacher forcing (Williams & Zipser, 1989). The combination of these factors exacerbates the problem of exposure bias (Ranzato et al., 2015), whereby the model does not learn how to distribute probability mass over sequences that are valid but different to the ground truth.

Recent work has investigated training the models directly on a performance based objective, either by optimising for BLEU score (Kumar et al., 2018a) or other quality metrics (Yuan et al., 2017). There is an implicit assumption that these metrics are in fact good proxies for question quality. We perform fine tuning using a range of rewards, including an adversarial objective. We show that although this leads to increases in those scores, the resulting models perform worse when evaluated by human workers, and the generated questions exploit weaknesses in the reward models.

## 2 RELATED WORK

Neural machine translation (NMT) has led to a number of advances in sequence modelling (Sutskever et al., 2014; Bahdanau et al., 2014; Gulcehre et al., 2016; Gehring et al., 2017), but for NMT the input and output sequences often have comparable lengths (Shi et al., 2016). This means that the attention weights correspond closely to per-token alignments, and the task becomes closer to a context-aware per-token transformation. Question generation requires identifying the important sections of the context before reordering some phrases and constructing some new ones, and so poses somewhat different challenges.

Summarisation involves taking a context document as input and generating a summary that should be considerably shorter, and is therefore more similar to the task of question generation. Cheng & Lapata (2016) propose an extractive neural summarisation framework that makes use of a Seq2Seq architecture with an attention mechanism, and Gu et al. (2016a) extend this by adding a pointer network. Nallapati et al. (2016a;b) present an abstractive model, and See et al. (2017) again extend these approaches by adding a pointer network. Paulus et al. (2017) propose a framework for fine tuning a summarisation model using reinforcement learning.

Early systems for generating questions were generally based around the use of some sort of templating or rule-based reordering of the context tokens (Heilman & Smith, 2010; Heilman, 2011; Agarwal et al., 2011; Ali et al., 2010; Danon & Last, 2017; Popowich & Winne, 2013; Chali & Golestanirad, 2016; Labutov et al., 2015; Mazidi & Nielsen, 2014).

Similar to these template based approaches, neural techniques have been used to generate questions from entities and relations in a knowledge base (Serban et al., 2016; Indurthi et al., 2017), but these require knowledge of the relations in advance and do not work from the raw textual input.

AQ systems can be used to augment datasets for training QA models, and Wang et al. (2017) and Tang et al. (2017) approach the task with this in mind. They generate questions using a Seq2Seq model, but primarily focus on the resulting improvement to the QA model. Yang et al. (2017) take a similar approach, using an AQ model to facilitate semi-supervised training of a QA model on a range of different domains.

The AQ task has also been approached by using only the document to generate questions, without conditioning on a specific answer. Subramanian et al. (2017) used named entity recognition to identify key phrases in the context before feeding this reduced input to a Seq2Seq model. They report an improved rating by human evaluators, but do not give any automated evaluation metrics for the generated questions. Kumar et al. (2018b) use a similar two-stage approach, adding attention and a pointer network to the decoder. Kumar et al. (2018a) further update this model by performing secondary policy gradient training, using BLEU and other automatic metrics as the rewards.

Du et al. (2017) use a Seq2Seq based model to generate questions conditioned on context-answer pairs, and build on this work by preprocessing the context to resolve coreferences and adding a pointer network (Du & Cardie, 2018). Similarly, Zhou et al. (2018) use a part-of-speech tagger to augment the embedding vectors. Both works perform a human evaluation of their models, and show significant improvement over their baseline. Song et al. (2018) use a modified context encoder based on multi-perspective context matching (Wang et al., 2016), similar to cross attention. Bahuleyan et al. (2017) used a variational encoder to generate multiple questions from a single context sentence. Gao et al. (2018) propose splitting the training data by the difficulty of question, and including this difficulty as part of the conditioning on the decoder.

Yuan et al. (2017) describe a Seq2Seq model with attention and a pointer network, with an additional encoding layer for the answer. They also describe a method for further tuning their model on a language model and question answering reward objective using policy gradient, but unfortunately do not perform any human evaluation.

## 3 MODEL DESCRIPTION

### 3.1 TASK DEFINITION

The training data consists of context-question-answer triples $(\mathbf{D}, \mathbf{Q}, \mathbf{A})$, that have been tokenised such that $\mathbf{D} = \{d_1, d_2, \ldots, d_{|D|}\}$ where $|D|$ is the number of tokens in the document, and similarly for the question and answer.

The task is to generate a natural language question $\hat{Y}$, conditioned on a document $\mathbf{D}$ and answer $\mathbf{A}$, by sampling from the parameterised conditional distribution at each time step given by $p(\hat{y}_t) = p_\theta(\hat{y}_t | \hat{y}_{<t}, \mathbf{D}, \mathbf{A})$. For example, given the input document $\mathbf{D} = $"this paper investigates assumptions in question generation" and $\mathbf{A} = $"question generation", the model should produce a question such as $\hat{Y} = $"what is investigated in the paper ?".

### 3.2 ENCODER

The context tokens are transformed into an embedding representation $\mathbf{d}_t$, by looking up the relevant entry in the word embedding matrix. We initialise with vectors from GloVe (Pennington et al., 2014) if the word exists in the GloVe vocabulary; otherwise we initialise with a random vector (Glorot & Bengio, 2010). We limit these embeddings to the top 2000 words in the SQuAD contexts and questions combined, ranked by frequency. Context words not in this vocabulary are mapped to the out-of-vocabulary token for the purpose of embedding.

The embedded tokens $\mathbf{d}_t$ are augmented with an additional binary feature, indicating whether that token comprises part of the answer or not, so that $\tilde{\mathbf{d}}_t = [\mathbf{d}_t; \mathbb{I}(d_t \in \mathbf{A})]$. The sequence of augmented embeddings is passed through a bidirectional LSTM layer (Hochreiter & Schmidhuber, 1997) to generate the context encodings $\mathbf{h}_t^d$.

For the model proposed by Yuan et al. (2017) with additional condition encoding, the encodings at the time steps corresponding to the answer span are concatenated with the word embeddings of these tokens, and this (shorter) sequence is passed through a second bidirectional LSTM layer. The condition encoding vector $\mathbf{h}^a$ is given by the RNN output at the last time step.

For the standard Seq2Seq model, the condition encoding is calculated as the mean context encoding of the answer span, $\mathbf{h}_a = \frac{1}{|A|} \sum_{t \in \mathbf{A}} \mathbf{h}_t^d$.

### 3.3 DECODER

At each time step, a weighted context vector $\mathbf{v}_t$ is computed using an attention mechanism to calculate soft alignments with the context document, and taking a weighted sum of the context encodings. This vector $\mathbf{v}_t = \sum_{t'}^{|D|} \alpha_{t,t'} \mathbf{h}_{t'}^d$ is used as the input to a unidirectional LSTM layer, giving the outputs $\mathbf{o}_t$.

The initial state for the decoder RNN is calculated according to $\mathbf{s}_0 = \tanh(\mathbf{W}_0\mathbf{r} + \mathbf{b}_0)$ where $\mathbf{r} = \mathbf{L}\mathbf{h}^a + \frac{1}{n}\sum_t^{|D|} \mathbf{h}_t^d$, and $\mathbf{L}, \mathbf{W}_0$, and $\mathbf{b}_0$ are learned parameters.

Alignment scores $\alpha_t$ are calculated using an attention mechanism (Bahdanau et al., 2014). Briefly, this takes the form of a fully connected network with a single hidden layer and a softmax output layer, that takes the current context encoding and previous hidden state as inputs, and produces a distribution over context time steps $\alpha_{t,i} = \frac{\exp(e_{t,i})}{\sum_j \exp(e_{t,j})}$ as output, where $\mathbf{e}_t = f(v_t, s_{t-1})$ is a fully connected neural network with $\tanh$ activation.

### 3.4 COPY MECHANISM

In order to handle unknown words, the model is able to generate tokens from two vocabularies: a *shortlist* vocabulary $\mathcal{V}_s$ of common context-independent words, and the *location* or copy vocabulary $\mathcal{V}_c$ formed of the words in the context document, indexed by their location in the context.

To calculate the distribution over shortlist tokens $\mathbf{p}_t^s$, the output of the LSTM cell at each step is projected into the dimensionality of the shortlist vocabulary, and normalised with a softmax activation, so that $\mathbf{p}_t^s = \text{softmax}(\mathbf{W}_s\mathbf{o}_t + b_s)$ where $\mathbf{W}_s, b_s$ are learned parameters and $\mathbf{o}_t$ is the output of the LSTM at each time step.

The distribution over context locations for the pointer network $\mathbf{p}_t^c$ is calculated by reusing the alignments from the attention mechanism, giving $\mathbf{p}_t^c = \alpha_t$.

The combined distribution is then calculated by concatenating the shortlist and location distributions, weighted by a switch variable that controls the degree of mixing of the two distributions. This switch variable $z_t$ is calculated at each step by passing $\mathbf{v}_t$ and $\mathbf{y}_{t-1}$ as inputs to a feedforward network with two hidden layers and $\tanh$ activation, with a single output variable passed through a sigmoid activation, $z_t = \sigma(f(\mathbf{v}_t, \mathbf{y}_{t-1}))$. The final output distribution over shortlist and location vocabularies is therefore given by $\mathbf{p}_t = [z_t\mathbf{p}_t^s; \ (1 - z_t)\mathbf{p}_t^c]$, where $[]$ is used to denote concatenation.

### 3.5 TRAINING

The ground truth data was encoded as a sequence of one-hot vectors $\tilde{\mathbf{q}}(t)$ over the combined shortlist and location vocabularies, with $\tilde{q}_i(t) = \mathbb{I}(w_i = q_t)$ for $w_i \in \mathcal{V}_s$. Tokens that did not occur either in the shortlist or context were encoded as an out-of-vocabulary token.

We trained the model on a maximum likelihood objective with teacher forcing Williams & Zipser (1989), using the Adam (Kingma & Ba, 2014) optimisation algorithm, and perform early stopping based on the negative log-likelihood of the development set. Dropout (Srivastava et al., 2014) and variational dropout (Gal & Ghahramani, 2015) were used where appropriate. The official SQuAD test set is not public, and we use the split published by Du et al. (2017) for our experiments.

For inference, we used beam search (Graves, 2012) with a width of 32. We also zeroed out the probability mass for the out-of-vocabulary token, to force the decoder to generate valid words.

## 4 EXPERIMENTS

### 4.1 COPY MECHANISM FORMULATION

Using a copy mechanism allows the model to generate language from a mixture of two vocabularies: a pre-defined *shortlist* of common words, and a context specific *location* vocabulary of words that appear in the source document. The probability of generating a token from one of these vocabularies compared to the other is controlled by the switch variable $z_t$.

For training samples where the shortlist and location vocabularies do not overlap, the correct value of the switch variable can easily be inferred, since there is only one way to generate each word. In practice this is rarely the case: the vocabularies often overlap significantly, and there may also be repetition *within* the context.

The original use of pointer networks for NLP (Gulcehre et al., 2016) was in the context of NMT, where the source and target language are different and the vocabularies can be assumed to be orthogonal, except for words which are named entities and must therefore be copied. Gulcehre et al. (2016) therefore assumed that words are generated from the shortlist by default, except when they must be copied. For the case where there is a choice of copy location, they simply selected the earliest, and we consider this approach as our default model.

CopyNet, concurrently proposed by Gu et al. (2016b), outputs a mixture of logits for the shortlist and location vocabularies, takes a softmax over these logits, and subsequently sums the probability mass for the overlapping tokens. This effectively makes the switch variable and choice of location into latent variables. See et al. (2017) and Du & Cardie (2018) also combine the probabilities for tokens with the same value. Yuan et al. (2017) use a copy mechanism as part of their model, but do not explicitly discuss the overlapping vocabulary problem and we assume that they treat all shortlist and location tokens as orthogonal. In each of these cases, the design choices are somewhat arbitrary, and were not tested.

There are two ways in which the vocabularies may overlap: within the location vocabulary, and between the location and shortlist vocabulary. We can remove a source of bias from the model by making the choice of copy location latent, by summing the probabilities of generating a token from each possible location, so that $p_t'^c(w) = \sum_i p_t^c(w_i)\mathbb{I}(w_i = w)$ where $i$ is an index on the locations in the context. We can also treat the switch variable as latent by summing over vocabularies, giving $p_t'(w) = z_t\, p_t^s(w) + (1 - z_t)p_t^c(w)$ for $w \in \mathcal{V}_s \cap \mathcal{V}_c$ and $p_t'(w) = p_t(w)$ otherwise.

It is not always clear that removing bias from a model will improve performance, and it may be better to use of our understanding of the problem to guide the model. We expect that the question should be similar to the language found near the answer span, and so design a heuristic detailed in Appendix A to incorporate this prior belief. The heuristic seeks to bias the model during training to copy contiguous runs of tokens from close to the answer span.

## 4.2 FINE TUNING

Generated questions should be formed of language that is both *fluent* and *relevant* to the context and answer. We experiment with fine tuning a trained model using rewards given by the negative perplexity under a LSTM language model and the F1 score attained by a question answering system, as well as a weighted combination of both. The language model is a standard recurrent neural network formed of a single LSTM layer. For the QA system, we use QANet (Yu et al., 2018) as implemented by Kim (2018).

Additionally, we propose a novel approach by learning the reward directly from the training data, using a *discriminator* detailed in Appendix B. We pre-trained the discriminator to predict whether an input question and associated context-answer pair was generated by our model, or originated from the training data. We also interleaved updates to the discriminator within the fine tuning phase, allowing the discriminator to become adversarial and adapt alongside the generator.

These rewards $R(\hat{Y})$ were used to update the model parameters via the REINFORCE policy gradient algorithm (Williams, 1992), according to $\nabla\mathcal{L} = \nabla\frac{1}{l}\sum_t(\frac{R(\hat{Y})-\mu_R}{\sigma_R})\log p(\hat{y}_t|\hat{y}_{<t}, \mathbf{D}, \mathbf{A})$. We teacher forced the decoder with the generated sequence to reproduce the activations calculated during beam search to enable backpropagation. All rewards were normalised with a simple form of PopArt (Hasselt et al., 2016), with the running mean $\mu_R$ and standard deviation $\sigma_R$ updated online during training. We continued to apply a maximum likelihood training objective during this fine tuning.

## 4.3 EVALUATION METRICS

We report the negative log-likelihood (NLL) of the test set under the model, as well as the corpus level BLEU-4 score (Papineni et al., 2002) of the generated questions compared to the ground truth. We report the macro-averaged F1 score attained by a QA system, which can be viewed as a form of reconstruction score, since it should be possible to recover the answer used to generate a good question. We also report the perplexity of generated questions under a LSTM language model (LM) trained on the questions from SQuAD.

# 5 RESULTS

## 5.1 AUTOMATIC METRICS

| Features | | | | Metrics | | | |
|:---:|:---:|:---:|:---:|:---:|:---:|:---:|:---:|
| Smart Heuristic | Latent Switch | Latent Location | Additional Encoding | NLL | BLEU | QA | LM |
| - | - | - | - | 43.7 | 11.4 | 69.1 | 61.3 |
| ✓ | - | - | - | 43.1 | 11.9 | 69.5 | 60.7 |
| - | ✓ | - | - | 41.3 | 12.3 | 70.5 | 59.5 |
| - | - | ✓ | - | 42.7 | 11.8 | 70.5 | 66.3 |
| - | ✓ | ✓ | - | 40.5 | 12.9 | 71.1 | 63.4 |
| - | - | - | ✓ | 43.0 | 13.1 | 71.4 | 54.0 |
| ✓ | - | - | ✓ | 43.3 | 12.9 | 71.5 | 57.2 |
| - | ✓ | - | ✓ | 40.6 | **13.1** | **72.6** | 55.0 |
| - | - | ✓ | ✓ | 42.7 | 12.4 | 71.8 | 60.9 |
| - | ✓ | ✓ | ✓ | **39.5** | 12.6 | 71.0 | **49.5** |
| Ground truth | | | | - | - | 71.2 | 101.5 |

Table 1: Automatic evaluation metrics evaluated for various formulations of the copy mechanism. QA and LM refer to the QA F1 score and language model perplexity scores attained by generated questions. The bottom row shows the performance of the QA and language models on the ground truth data.

Table 1 shows the values of the automatic metrics for various configurations of the copy mechanism, for both a standard Seq2Seq condition encoding and for the additional encoding used by Yuan et al. (2017). The LM perplexity of generated questions is lower than the ground truth for all configurations; this is to be expected, since the question generator is itself effectively a conditional language model.

For the standard Seq2Seq condition encoding, using a smarter copy location heuristic to bias the model during training leads to a small improvement of +0.5 BLEU. Allowing the model to instead learn this heuristic by making both the copy location and the switch variable latent leads to significant improvements of +1.5 BLEU and +2.0 QA score.

When the additional condition encoding is included, we no longer observe significant improvement for the latent formulations of the copy mechanism, and instead find that performance stays the same or decreases by up to 0.7 BLEU. The additional encoding layer increases the number of parameters in the model; when this is coupled with the additional freedom in the copy mechanism, the model is unable to learn as effectively.

Table 2 shows the changes in automatic metrics after fine tuning using various external rewards. Optimising for QA, LM and discriminator rewards improves those scores, although a larger improvement in LM score is achieved with a combined QA and LM reward. The biggest improvement in discriminator score is achieved using an adversarial objective, and using a weighted sum of all three objectives leads to improvements in all three rewards. The BLEU score decreases in all cases, as the rewards are not coupled to the training data.

## 5.2 HUMAN EVALUATION

We follow the standard approach in evaluating machine translation systems (Koehn & Monz, 2006), as used for AQ by Du & Cardie (2018). We asked three workers to rate 300 generated questions between 1 (poor) and 5 (good) on two separate criteria: the fluency of the language used, and the relevance of the question to the context document and answer.

| Features | | | | Metrics | | | | |
|---|---|---|---|---|---|---|---|---|
| QA reward | LM reward | Discriminator reward | Adversarial discriminator | NLL | BLEU | QA | LM | Discriminator |
| - | ✓ | - | - | -0.7 | -1.9 | -3.7 | -13.4 | +1.5 |
| ✓ | - | - | - | +1.7 | -4.5 | **+3.9** | +226 | +5.4 |
| ✓ | ✓ | - | - | -0.5 | -2.6 | +2.0 | **-16.3** | +2.9 |
| - | - | ✓ | - | **-0.8** | -1.8 | -2.1 | -9.4 | +2.5 |
| - | - | ✓ | ✓ | +6.4 | -2.7 | -2.5 | -1.0 | **+10.8** |
| ✓ | ✓ | ✓ | ✓ | +1.0 | -2.4 | +1.3 | -6.2 | +10.0 |

Table 2: Changes in automatic evaluation metrics after models were fine tuned on various objectives. The discriminator reward refers to the percentage of generated sequences that fooled the discriminator. Lower LM and NLL scores are better.

| Model | Fluency | Relevance |
|---|---|---|
| S2S +Copy | 3.34 | 3.12 |
| +Latent Switch +Latent Location | **3.51** | **3.42** |
| +QA, LM rewards | 3.05 | 2.75 |
| a +QA, LM, discriminator rewards +Adversarial discriminator | 2.89 | 2.82 |
| Ground Truth | 4.67 | 4.72 |

Table 3: Summary of human evaluation of selected models

The mean of the pairwise inter-rater Fleiss' Kappa (L. Fleiss, 1971) agreement metrics was 0.45 for fluency, and 0.44 for relevance, corresponding to moderate agreement. While this seems low, the metric considers the rating classes to be unordered and equally different from each other, and so is pessimistic.

As shown in Table 3, removing bias by making the copy mechanism handle overlapping words in a latent manner leads to improved fluency and relevance. The models that were fine tuned on external rewards achieve worse human scores. We note that although BLEU has been shown to be an imperfect metric (Paulus et al., 2017; Chaganty et al., 2018), in this instance it is sufficient to predict the human ranking of the different models.

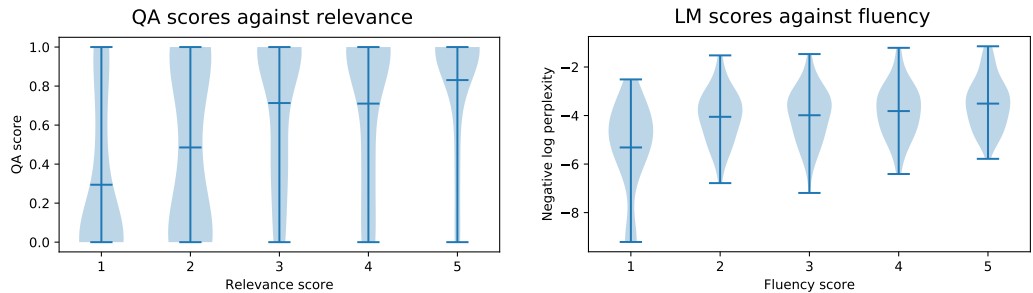

(a) QA scores plotted against human relevance scores for all rated questions.

(b) LM scores plotted against human fluency scores for all rated questions.

Figure 1: Comparison of human and automatic metrics.

Figure 1 shows the automatic scores against human ratings for all rated questions. The correlation coefficient between human relevance and automatic QA scores was 0.439, and between fluency and LM score was only 0.355. While the automatic scores are good indicators of whether a question will achieve the lowest human rating or not, they do not differentiate clearly between the higher

ratings: training a model on these objectives will not necessarily learn to generate better questions. A good question will likely attain a high QA and LM score, but the inverse is not true; a sequence may exploit the weaknesses of the metrics and achieve a high score *despite* being unintelligible to a human. Using a weighted combination of rewards is not sufficient to provide a useful training objective.

Table 4 shows examples of generated questions for the fine tuned models. Training on a QA reward has caused the model to learn to exploit this reward, by simply using a few keywords to point at the answer. This suggests an alternative application of AQ models for generating adversarial data for QA systems and exposing their failure cases, similar to the work by Jia & Liang (2017).

Training on an adversarial objective should prevent the generator from being able to exploit the weaknesses of the reward model. We find that although the generated sequences appear reasonable, the model fine tuned on an adversarial reward was not more highly rated by the human workers.

| **Context** |
| --- |
| although united methodist practices and interpretation of beliefs have evolved over time , these practices and beliefs can be traced to the writings of the church 's founders , especially **john wesley and charles wesley** ( anglicans ) , but also philip william otterbein and martin boehm ( united brethren ) , and jacob albright ( evangelical association ) . |
| **Answer** |
| john wesley and charles wesley |
| **Ground Truth Question** |
| who were two of the founders of the united methodist church ? |
| **No fine tuning** |
| which two methodist can be traced to the church 's founders ? |
| **LM reward** |
| according to the writings of the church 's founders , according to the writings of the church 's founders , [...] |
| **QA reward** |
| who in anglicans ? |
| **LM and QA reward** |
| who are the writings of the church 's founders ? |
| **Discriminator reward** |
| who founded the church 's founders ? |
| **Discriminator reward, adversarial discriminator** |
| who were two western methodist practices ? |
| **LM, QA and discriminator reward, adversarial discriminator** |
| who are the anglicans of the church ? |

Table 4: Example generated questions for various fine-tuning objectives. The model trained on a QA reward has learned to simply point at the answer and exploit the QA model, while the model trained on a language model objective has learned to repeat common phrase templates.

## 6 CONCLUSION

In this paper we clarify two fundamental assumptions in recent work on question generation. We show that, for standard Seq2Seq models, removing inductive bias by making the source of non-unique words latent improves the quality of generated questions. We perform a human evaluation to confirm these findings.

We also find that although policy gradient methods can be used to optimise for external rewards, these rewards do not correlate well with question quality despite being intuitively good choices. The generator may simply learn to exploit the weaknesses of the reward model, suggesting a possible use of AQ systems to generate adversarial training data for those reward models. Fine tuning on an adversarial objective did not lead to improved question quality.

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

## A   SMART COPY HEURISTIC

---
**Algorithm 1** Smart copy heuristic

---
**if** the token exists only once in the context **then**
    use that location
**else**
    **if** there are multiple instances
    **and** the previously encoded token was from the location vocabulary
    **and** the token at the next position in the context is the correct token **then**
        use that position
    **else**
        find the occurrence nearest the answer span

---

## B   DISCRIMINATOR ARCHITECTURE

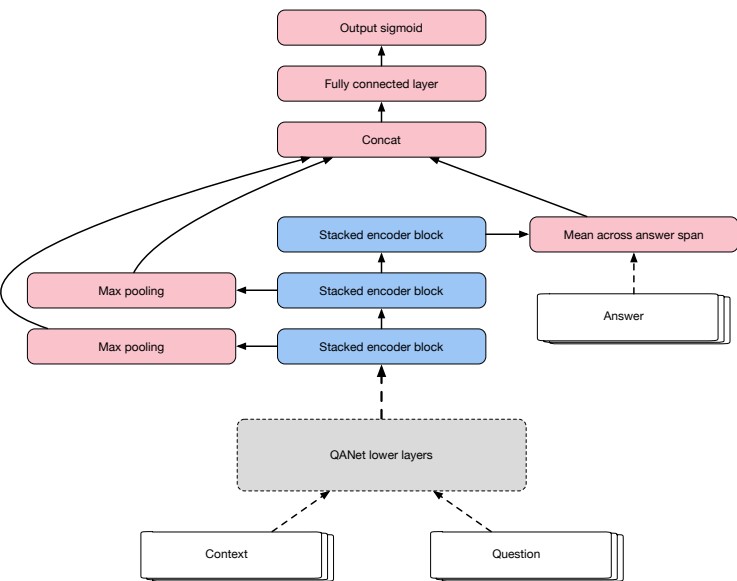

Figure 2: Discriminator architecture diagram.

We used an architecture based on a modified QANet as shown in Figure 2, replacing the output layers of the model to produce a single probability. Since the discriminator is also able to consider a full context-question-answer triple as input (as opposed to a context-question pair for the QA task), we fused this information in the output layers.

Specifically, we applied max pooling over time to the output of the first two encoders, and we took the mean of the outputs of the third encoder that formed part of the answer span. These three reduced encodings were concatenated, a 64 unit hidden layer with ReLU activation applied, and the output passed through a single unit sigmoid output layer to give the estimated probability that an input context-question-answer triple originated from the ground truth dataset or was generated.

## C  HYPERPARAMETER VALUES

| Parameter | Value |
|---|---|
| Learning rate | $2 \times 10^{-4}$ |
| Embedding dimension | 200 |
| RNN hidden units | 768 |
| Shortlist vocab size | 2000 |
| Dropout rate | 0.3 |
| Beam width | 32 |

Table 5: Hyperparameter values used for AQ models

| Parameter | Value |
|---|---|
| Vocab size | $20,000$ |
| Embedding dimension | 200 |
| RNN hidden units | 384 |
| Dropout rate | 0.3 |

Table 6: Hyperparameter values used for language model

| Model type | LM weight | QA weight | Discriminator weight |
|---|---|---|---|
| LM | 0.1 | - | - |
| QA | - | 1.0 | - |
| LM+QA | 0.25 | 0.5 | - |
| Disc | - | - | 1.0 |
| Disc+Adversarial | - | - | 1.0 |
| LM+QA+Disc | 0.25 | 0.5 | 0.5 |

Table 7: Model specific reward weights

