# OpenReview forum: "Assumption Questioning: Latent Copying and Reward Exploitation in Question Generation"
_ICLR.cc/2019/Conference_

### Official Review · AnonReviewer3 · 2018-11-02
**Novelty limited and experiments not convincing enough**

**Rating:** 5
**Confidence:** 4

**Review:**

In the paper, author investigate the use of copy mechanisms for the question generation task. It evaluates on the SQuAD dataset. The model is a popular seq2seq/encoder-decoder model with copy mechanisms using pointer networks.

Pros:
It is well motivated. For the question generation task, a word to be predicted can be from either a global vocabulary list or copied from the given documents (location vocabulary).  There are some overlap between these two vocabulary lists.  This paper mainly investigates this issue.

It is well written and easy to follow.

Interesting analysis of human/automatic metrics.

Cons:
The tricks here are a bit of ad hoc. It is better to have a systemic study.

Baseline results are too low. E.g., officially QANet results (from the paper) on SQuAD v1 is around 82.7 (my implementation obtains 83.1). However in the paper, its best result is 72.6 in terms of F1 score.

The authors only evaluated on one dataset. It is hard to convincing.

It is lack of comparison results of question generation in literature.

---

> ### Author Response · Authors · 2018-11-16
> **Response to review #3**
>
> Thanks for taking the time to review our paper!
>
> > The tricks here are a bit of ad hoc. It is better to have a systemic study.
>
> We would argue somewhat the opposite - the existing implementations of copy mechanisms make arbitrary choices regarding dealing with overlapping vocabularies, whereas we aim to test all possible interpretations systematically using the same underlying model.
>
> > Baseline results are too low. E.g., officially QANet results (from the paper) on SQuAD v1 is around 82.7 (my implementation obtains 83.1). However in the paper, its best result is 72.6 in terms of F1 score.
>
> Note that the 72.6 score refers to the score achieved using _generated questions_, not the ground truth questions - it is therefore an indication of the quality of the questions and not of the QA model.
>
> > The authors only evaluated on one dataset. It is hard to convincing.
>
> This is a valid concern. Unfortunately there is no standard AQ dataset, and many of the more interesting QA datasets do not guarantee that the answer is present within the context document, meaning that a different approach must be used. That said, the recent paper "Paragraph-level Neural Question Generation
> with Maxout Pointer and Gated Self-attention Networks" from Zhao et al used a subset of the MSMARCO dataset, and we will aim to investigate the models on this dataset in future.
>
> > It is lack of comparison results of question generation in literature.
>
> This is also a valid concern. While the intention was to investigate the relative effects of various formulations of an increasingly popular model component, we should do better to a) make this clear and b) place the paper more clearly within the wider context of results.

---

### Official Review · AnonReviewer1 · 2018-11-03
**Studied the problem of question generation. However the paper is hard to follow and the proposed model lacks novelty.**

**Rating:** 3
**Confidence:** 4

**Review:**

The paper studies question generation, which is an important problem in many real applications. The authors propose to use better caching model and more evalution methods to deal with the problem. However, the paper is poorly written and hard to follow, and the proposed model lacks of novelty. The main reasons are as below:

1) In model section, the task definition is not clear. It is expected to see what's the question generation task studied in this paper. An example or a model overview will definitly help.

2) The encoder and decoder are not novel, it is expected to cite and compare with the existing similar encoder architecture, such as the encoder proposed in bidaf "Seo, Minjoon, et al. "Bidirectional attention flow for machine comprehension." arXiv preprint arXiv:1611.01603 (2016)."  The math symbols are aligned, for example, h_a or h^a is used to represent the encoding. Besides, adding the binary feature in the embedding is not necessary, the LSTM model could learn such sequential correlation. The decoder description is not clear as well and expected to compare with existing work (e.g., bidaf) to show the difference.

3) The proposed copy mecahnism is not clear. A formal definition of s_t, v_t and y_(t-1) should be given before defining the p_t. A more serious question, what is the fuse operation used to define p_t? concat, elementwise_plus or others?

4) In the training, how to deal the ground truth that are not in the vocab? The authors stated "using a modified heuristic described below", but no follow-ups in the paper.

5) The paper is not well written and organized. Small typos: in introduction, 'and and answer span', 'and output and output sequences'. In model, 'Glorot initialization', 'Bahdanau attention', it is not the common way to cite others' work. In encoder, the defintion of the state for decoder could be reorganized to the decoder.

I have read the authors' detailed rebuttal. Thanks.

---

> ### Author Response · Authors · 2018-11-16
> **Response to review #1**
>
> Thanks for taking the time to review our paper!
>
>
> > 1) In model section, the task definition is not clear. It is expected to see what's the question generation task studied in this paper. An example or a model overview will definitly help.
>
> This is a valid concern. We will update the paper with a clearer task defintion and a simple example to illustrate it.
>
> > 2) The encoder and decoder are not novel, it is expected to cite and compare with the existing similar encoder architecture, such as the encoder proposed in bidaf "Seo, Minjoon, et al. "Bidirectional attention flow for machine comprehension." arXiv preprint arXiv:1611.01603 (2016)."  The math symbols are aligned, for example, h_a or h^a is used to represent the encoding. Besides, adding the binary feature in the embedding is not necessary, the LSTM model could learn such sequential correlation. The decoder description is not clear as well and expected to compare with existing work (e.g., bidaf) to show the difference.
>
> Note that we do not claim to present a novel encoder or decoder, but rather review a popular component that does not have a standardised formulation and that involves a number of design choices that have never been compared.
>
> BiDAF is not an appropriate comparison in this case. Firstly, the task is different (it is a QA model, not AQ). Secondly, it fuses information from a context and a question, where the question is *not* a subspan of the context - in our case, the answer *is* a span within the context. Thirdly, BiDAF merely points to the correct answer span within the context and does not involve a decoding step.
>
> > 3) The proposed copy mecahnism is not clear. A formal definition of s_t, v_t and y_(t-1) should be given before defining the p_t. A more serious question, what is the fuse operation used to define p_t? concat, elementwise_plus or others?
>
> We will update the paper to ensure that the relevant symbols and operations are defined. [p^s; p^c] in this case means concatenation.
>
> > 4) In the training, how to deal the ground truth that are not in the vocab? The authors stated "using a modified heuristic described below", but no follow-ups in the paper.
>
> The heuristic described is regarding words that are not in the shortlist vocabulary, but are present in more than one location in the source context (in which case the choice of copy location must either be selected, or left as a latent variable). Ground truth tokens that are not present in either vocabulary (which is very rare) are encoded as the special OOV token as standard. We will update the paper to make this distinction more clear.
>
> > 5) The paper is not well written and organized. Small typos: in introduction, 'and and answer span', 'and output and output sequences'. In model, 'Glorot initialization', 'Bahdanau attention', it is not the common way to cite others' work. In encoder, the defintion of the state for decoder could be reorganized to the decoder.
>
> Thank you for bringing these typos to our attention. Likewise, Glorot and Bahdanau should be cited at those points.

---

### Official Review · AnonReviewer2 · 2018-11-06
**Some mixed results and needs further analysis**

**Rating:** 4
**Confidence:** 4

**Review:**

This paper presents question generation models by designing variations of copying mechanism and reward functions. Experimental results show that different copying mechanism can improve upon basic seq2seq models, some of the reward functions also produce better results. I think the results are interesting, especially the ones compared with human evaluation (fig. 1), but it's might be better to explain on which aspect each of the feature contributes to the improvement. For instance, the authors can give some insights based on empirical results on what kind of questions will benefit from each type of copying.


The authors should better organize table 1 and 2, and inform the readers on what is the consistent conclusion (if any). For table 2, there is no result for adding "adversarial discriminator" only. Also the item "+226" on the second row, is that an error?

---

> ### Author Response · Authors · 2018-11-16
> **Response to Review #2**
>
> Thanks for taking the time to review our paper!
>
> > The authors should better organize table 1 and 2, and inform the readers on what is the consistent conclusion (if any).
>
> Is something in particular that is unclear regarding the tables? We do discuss the overall conclusion but will check again that this is clear.
>
>
> > For table 2, there is no result for adding "adversarial discriminator" only.
>
> This combination is not possible - the discriminator must be used a reward in order to also be adversarial. We will look into changing this label to make this more clear.
>
> > Also the item "+226" on the second row, is that an error?
>
> This is not an error - as discussed, training on a QA objective leads to significant decreases in the LM score, as the model simply learns to exploit the QA reward.

---

### Meta-Review · Area_Chair1 · 2018-12-14
**relatively weak contributions and novelty**

**Confidence:** 5
**Recommendation:** Reject

**Metareview:**

This paper investigates copying mechanisms and reward functions in sequence to sequence models for question generation. The key findings are threefold: (1) when the alignments between input and output are weak, it is better to use latent copying mechanism to soften the model bias toward copying, (2) while policy gradient methods might be able to improve automatic scores, their results poorly align with human evaludation, and (3) the use of adversarial objective also does not lead to useful training signals.

Pros:
The task is well motivated and the paper presents potentially useful negative results on policy gradient and adversarial training.

Cons:
All reviewers found the clarity and organization of the paper requires improvements. Also, the proposed methods are reletively incremental and the empirical results are not strong. While the rebuttal answered some of the clarification questions, it does not address major concerns about the novelty and contributions.

Verdict:
Reject due to relatively weak contributions and novelty.